# Intestinal Permeability and Dysbiosis in Female Patients with Recurrent Cystitis: A Pilot Study

**DOI:** 10.3390/jpm12061005

**Published:** 2022-06-20

**Authors:** Cristina Graziani, Lucrezia Laterza, Claudia Talocco, Marco Pizzoferrato, Nicoletta Di Simone, Silvia D’Ippolito, Caterina Ricci, Jacopo Gervasoni, Silvia Persichilli, Federica Del Chierico, Valeria Marzano, Stefano Levi Mortera, Aniello Primiano, Andrea Poscia, Francesca Romana Ponziani, Lorenza Putignani, Andrea Urbani, Valentina Petito, Federica Di Vincenzo, Letizia Masi, Loris Riccardo Lopetuso, Giovanni Cammarota, Daniela Romualdi, Antonio Lanzone, Antonio Gasbarrini, Franco Scaldaferri

**Affiliations:** 1CEMAD Digestive Diseases Center, Fondazione Policlinico Universitario “A. Gemelli” IRCCS, Università Cattolica del Sacro Cuore, 00168 Rome, Italy; cristina.graziani@guest.policlinicogemelli.it (C.G.); marco.pizzoferrato@policlinicogemelli.it (M.P.); francescaromana.ponziani@policlinicogemelli.it (F.R.P.); valentina.petito@policlinicogemelli.it (V.P.); federica.divincenzo30@gmail.com (F.D.V.); letizia.masi94@gmail.com (L.M.); lorisriccardo.lopetuso@policlinicogemelli.it (L.R.L.); giovanni.cammarota@policlinicogemelli.it (G.C.); antonio.gasbarrini@unicatt.it (A.G.); franco.scaldaferri@unicatt.it (F.S.); 2Dipartimento di Medicina e Chirurgia Traslazionale, Università Cattolica del Sacro Cuore, 00168 Rome, Italy; claudia.talocco@yahoo.it; 3Department of Biomedical Sciences, Humanitas University, Via Rita Levi Montalcini 4, 20072 Pieve Emanuele, Italy; nicoletta.disimone@hunimed.eu; 4IRCCS Humanitas Research Hospital, Via Manzoni 56, 20089 Rozzano, Italy; 5Ginecologia ed Ostetricia, Dipartimento di Scienze Della Vita e Sanità Pubblica, Fondazione Policlinico Universitario “A. Gemelli” IRCCS, 00168 Rome, Italy; silvia.dippolito@policlinicogemelli.it (S.D.); caterina.ricci@policlinicogemelli.it (C.R.); daniela.romualdi@policlinicogemelli.it (D.R.); antonio.lanzone@policlinicogemelli.it (A.L.); 6UOC Chimica, Biochimica e Biologia Molecolare Clinica, Dipartimento di Scienze Biotecnologiche di Base, Cliniche Intensivologiche e Perioperatorie, Fondazione Policlinico Universitario “A. Gemelli” IRCCS, 00168 Rome, Italy; jacopo.gervasoni@policlinicogemelli.it (J.G.); silvia.persichilli@policlinicogemelli.it (S.P.); aniello.primiano@policlinicogemelli.it (A.P.); andrea.urbani@policlinicogemelli.it (A.U.); 7Multimodal Laboratory Medicine Research Area, Unit of Human Microbiome, Bambino Gesù Children’s Hospital, IRCCS, 00165 Rome, Italy; federica.delchierico@opbg.net (F.D.C.); valeria.marzano@opbg.net (V.M.); stefano.levimortera@opbg.net (S.L.M.); 8UOC ISP Prevention and Surveillance of Infectious and Chronic Diseases, Department of Prevention-Local Health Authority (AUSR-AV2), 60035 Jesi, Italy; andrea.poscia@sanita.marche.it or; 9Department of Diagnostic and Laboratory Medicine, Unit of Microbiology and Diagnostic Immunology, Unit of Microbiomics and Multimodal Laboratory Medicine Research Area, Unit of Human Microbiome, Bambino Gesù Children’s Hospital, IRCCS, 00146 Rome, Italy; lorenza.putignani@opbg.net

**Keywords:** intestinal permeability, gut microbiome, recurrent cystitis, dysbiosis

## Abstract

Recurrent cystitis (RC) is a common disease, especially in females. Anatomical, behavioral and genetic predisposing factors are associated with the ascending retrograde route, which often causes bladder infections. RC seems to be mainly caused by agents derived from the intestinal microbiota, and most frequently by *Escherichia coli*. Intestinal contiguity contributes to the etiopathogenesis of RC and an alteration in intestinal permeability could have a major role in RC. The aim of this pilot study is to assess gut microbiome dysbiosis and intestinal permeability in female patients with RC. Patients with RC (*n* = 16) were enrolled and compared with healthy female subjects (*n* = 15) and patients with chronic gastrointestinal (GI) disorders (*n* = 238). We calculated the Acute Cystitis Symptom Score/Urinary Tract Infection Symptom Assessment (ACSS/UTISA) and Gastrointestinal Symptom Rating Scale (GSRS) scores and evaluated intestinal permeability and the fecal microbiome in the first two cohorts. Patients with RC showed an increased prevalence of gastrointestinal symptoms compared with healthy controls. Of the patients with RC, 88% showed an increased intestinal permeability with reduced biodiversity of gut microbiota compared to healthy controls, and 68% of the RC patients had a final diagnosis of gastrointestinal disease. Similarly, GI patients reported a higher incidence of urinary symptoms with a diagnosis of RC in 20%. Gut barrier impairment seems to play a major role in the pathogenesis of RC. Further studies are necessary to elucidate the role of microbiota and intestinal permeability in urinary tract infections.

## 1. Introduction

Recurrent cystitis (RC) is defined as more than two episodes of bladder infection in a 6-month period or more than three episodes in a year. Generally, urinary tract infections (UTIs) are much more common in women than in men, involving over 50% of the female population, of which at least 20–30% develop a recurrence [1]. This increased incidence in women can be partially explained by the presence of anatomical, behavioral and genetic predisposing factors [2,3,4]. Most urinary tract pathogens consist of facultative Gram-negative anaerobic bacilli, common microorganisms of the intestinal microbiota, mainly *Escherichia coli*, but they also belong to other Enterobacteriaceae (such as *Klebsiella* spp. and *Proteus* spp.). However, even Gram-positive microorganisms, such as *Staphylococcus saprophyticus* and *Enterococcus faecalis,* can act as pathogens [5].

The main route of infection is the fecal–perineal–urethral route, also known as the ascending retrograde route, which consists of the colonization of the vaginal introitus and/or the urethral meatus by fecal microbiota-derived bacteria, and the consequent colonization of the bladder through the urethra [6]. Thus, the intestine could act as a reservoir of uropathogens and the cross-talk between the intestinal and urogenital microbiome, the “gut–bladder axis”, could play a major role in UTIs’ pathogenesis [7]. 

Changes in epithelial permeability may represent a novel mechanism for visceral organ crosstalk and it may explain the overlapping symptomology of painful bladder syndrome and irritable bowel syndrome (IBS) [8].

The pathophysiology of painful bladder syndrome (PBS) is poorly understood. However, there is evidence of female predominance and a high incidence of IBS in these patients: up to 30–50% of patients diagnosed with IBS show symptoms of PBS, while up to 40% of patients diagnosed with PBS show symptoms that meet the criteria for IBS. The hypothesis is that the cross-sensitization between the bladder and colon is due to altered permeability in one organ, which affects the other organ, but we do not know which one is the first. [9,10]. However, there is limited knowledge of the mechanisms that link these conditions. According to this hypothesis of cross-organ visceral communication between the colon and bladder, previous experiments in rodent models have shown that colonic irritation is capable of producing irregular urination patterns, such as early onset of urination and increased urethral sphincter activity in rats [11]. Furthermore, there is evidence that active colonic inflammation induces abnormalities in the detrusor-muscular contractility of the bladder [12], and can increase vascular permeability in the bladder of female rats [13]. Conversely, bladder irritation results in increased visceral sensitivity to colonic stimulation. The induction of permeability in the bladder induces increased permeability in the colon, and, on the other side, inflammation of the colon likewise induces permeability in the urinary bladder. These findings suggest that altered permeability has a key role in the visceral organ cross-talk [8]. 

Based on this rationale, it is possible to hypothesize a further route of colonization of the bladder by an anterograde route in RC, possibly by the transmigration of bacteria or bacterial fragments from the intestine, particularly in the presence of impaired permeability, as demonstrated in murine models [14]. However, little is known about the contribution of intestinal permeability in the pathogenesis of recurrent cystitis.

Intestinal homeostasis depends on the good health of the gut barrier, a complex defensive system capable of separating the intestinal contents from the host tissues, which regulates nutrient absorption and allows interactions between the resident microbiota and the local immune system [15]. The gut barrier is constituted and regulated by many factors, including, first of all, the intestinal microbiota itself, which could also influence the microbiota of nearby organs, the mucus layer, the integrity of epithelial cells and intercellular junctions, and the innate and adaptive immune system associated with the mucosa [16]. In this context, an important measure of barrier integrity is intestinal permeability, the property that allows the exchange of solutes and fluids between the lumen and the intestinal mucosa. The increase in intestinal permeability as a marker of gut barrier dysfunction has been implicated in the pathogenesis of many gastrointestinal and extra-gastrointestinal diseases [17]. However, little or nothing is known about the relationship between RC, dysbiosis and increased intestinal permeability. 

Therefore, the aim of this pilot study is to evaluate the possible relationship between impaired gut barrier function and RC, through the investigation of the prevalence of increased intestinal permeability and dysbiosis in a cohort of female patients with RC (primary endpoint) compared to healthy women. To explore the possible crosstalk between the gut and the urinary tract and to support the rationale of a bi-directional gut–bladder dysfunction, we also evaluated the prevalence of RC in a cohort of patients with chronic gastrointestinal disorders. 

## 2. Materials and Methods

### 2.1. Study Design and Patients

We recruited three cohorts of patients: the first cohort (cohort I) consisted of female patients, aged 18 and over, who reported at least two episodes of acute uncomplicated cystitis in the last 6 months or three episodes in the last year, and came to our attention at the Gynecology Unit of the Fondazione Policlinico Universitario “A. Gemelli” IRCCS Hospital. The exclusion criteria for this cohort were the presence of morpho-functional alterations of the genitourinary tract, the diagnosis of complicated acute or chronic cystitis, pregnancy and lactation. The second cohort (control group, cohort II) was composed of healthy female subjects from the age of 18, followed up by the Gynecology Unit of the Fondazione Policlinico Universitario “A. Gemelli” IRCCS Hospital for routine controls, without a history of recurrent cystitis or any gastroenterological symptoms/disorders, and in the same age range. The third cohort (cohort III) was composed of female patients who attended the General Gastroenterology and Breath Test Clinics of the Fondazione Policlinico Universitario “A. Gemelli” IRCCS Hospital for GI disorders. Enrolled patients did not undergo any therapy at the time of enrollment and execution of examinations and tests. In addition, patients were required not to change their eating habits. 

The subjects in cohort I and II performed the Intestinal Permeability Test, based on the lactulose/mannitol ratio and, at the same time, on the measurement of exhaled H_2_ (the Lactulose Breath Test) for the assessment of oro-cecal transit time and Small Intestinal Bacterial Overgrowth (SIBO). On the same day, they provided a fecal sample for metagenomic 16S ribosomal RNA (rRNA) analysis of the intestinal microbiome. 

Subjects belonging to all three study cohorts were administered validated questionnaires for self-evaluation of urological symptoms—the Acute Cystitis Symptom Score (ACSS) questionnaire [18] and the UTI Symptom Assessment (UTISA) questionnaire [19]—and for gastrointestinal symptoms, the structured Gastrointestinal Symptom Rating Scale (GSRS) questionnaire was administered [20].

Enrolled patients and controls had no history of alcohol or drug abuse and they were not current smokers. Subjects participating in the study did not refer to any particular restricted dietary regimen (i.e., vegetarian or a low FODMAP diet) and they were asked not to change their dietary habits and to avoid the use of antibiotics 15 days before their enrollment in the study and microbiota analysis. Patients with RC were also required to provide the results of the last urine culture, in order to collect data about the causative agent of the urinary infection.

The study was approved by the Ethical Committee of Fondazione Policlinico Universitario “A. Gemelli” IRCCS Hospital and all the subjects participating in the study provided written informed consent to the study (Protocol Number 0011046/21 of 03/24/2021).

### 2.2. Intestinal Permeability Test

We developed a new method to evaluate the intestinal permeability index, according to our preliminary results (unpublished data). It showed the comparability of data on exhaled gas obtained from a standard H_2_ Lactulose Breath Test compared to data obtained after the concomitant administration of lactulose and mannitol to perform a Lactulose/Mannitol urinary test. Thus, we developed a contemporary breath and urinary test that was able to provide combined information: the H_2_ Lactulose/Mannitol Breath Test (L/M BT), thanks to the administration of two sugars, instead of lactulose alone. This method also allows the simultaneous determination of both H_2_ and CH_4_ measurements, providing information on oro-cecal transit time and SIBO, and at the same time, the determination of urinary lactulose/mannitol urinary ratio, for the estimation of intestinal permeability. This test has already been demonstrated to provide reliable information on the alteration of intestinal permeability [16,21,22,23,24,25] and it is currently part of the clinical practice in our center. In this way, by performing a single, simple, non-invasive, sensitive, reliable and repeatable test, it is possible to not only obtain information relating to the functionality of the intestinal epithelium, but also relating to any alterations in intestinal transit time or to the presence of SIBO. The L/M BT is performed through serial sampling of gases exhaled by the patient, such as hydrogen, carbon dioxide and methane, and the subsequent analysis of their concentrations, measured in parts per million (ppm) by means of a dedicated gas chromatograph. After the appropriate preparation the day before the test and an overnight fast, 17 samples of exhaled gas were obtained at time 0 (T0) and after taking 5 gr of mannitol (powder) dissolved in 200 mL of water and 10 gr of pure lactulose (15 mL of syrup), at intervals of 15 min in the following 4 h. An increase of ≥20 ppm in hydrogen within 90 min was considered the cut-off value for the determination of SIBO, according to the North American consensus criteria [26]. Patients also provided a urine sample at baseline (T0), before taking lactulose and mannitol. Then, they collected urine samples in the following 6 h to allow the measurement of the two sugars. The lactulose/mannitol ratio was considered increased and therefore indicative of increased intestinal permeability, for values ≥0.030 [27].

#### 2.2.1. Chemicals and Reagents

Water and acetonitrile (LC-MS grade) was purchased from Merck (Merck KGaA, Darmstadt, Germany). Formic acid (98%, LC-MS grade) was purchased from Baker (Mallinckrodt Baker Italia, Milano, Italia) and D-Mannitol-1 ^13^C, 1-1-d_2_, Lactulose ^13^C_12_ and ammonium formate was purchased from Sigma-Aldrich (Merck KGaA, Darmstadt, Germany). Lactulose, mannitol and chlorhexidine were purchased from BioChemica (AppliChem GmbH, Darmstadt, Germany). Stock solutions of mannitol (4 g/L) and lactulose (4 g/L) were prepared in water and stored at −80 °C. Internal standards (IS) stock solutions containing 500 µg/mL D-mannitol-1 ^13^C.1, 1-d_2_ and lactulose ^13^C_12_ were prepared in water and stored at −80 °C. Working solutions were prepared in water/acetonitrile (20/80, *v*/*v*) at concentrations of 1600 µg/mL for mannitol and 640 µg/mL for lactulose. Serial dilutions from working solutions were used to prepare seven-point calibration curves for both mannitol and lactulose (10, 40, 80, 160, 320, 640 µg/mL; 2.5, 10, 20, 40, 80.,160 µg/mL, respectively) and kept at −20 °C until use. The calibration curve included a zero (only solvent) and a blank (solvent plus IS), which were not used for the construction of calibration curves. D-Mannitol-1 ^13^C, 1-1-d_2_ and lactulose ^13^C_12_ stock solutions were diluted with acetonitrile to achieve a final concentration of 5 µg/mL and 2.5 µg/mL for D-Mannitol-1 ^13^C, 1-1-d_2_ and lactulose ^13^C_12_, respectively. 

#### 2.2.2. Sample Collection and Treatment 

Urine samples were collected at two time points: T0 (at the start of the Lactulose/Mannitol Breath Test when the patient had fasted from solids and liquids for at least 8 h, before the assumption of the two sugars) and at T6 (for 6 hours after the consumption of the two sugar solutions: 10 gr of lactulose and 5 gr of mannitol). An aliquot of the urine sample collected at 6 h was taken for analysis. Then, 150 µL of chlorhexidine (1.9 gr/100 mL) was added to the urine samples as a preservative. Samples were stored at −80 °C until analysis.

Before the analysis for the L/M ratio, urine samples were left to thaw at room temperature, then stirred for 1 min using a vortex mixer and centrifuged at 5000× *g* for 4 min to remove the sediment according to the laboratory procedure. IS solution (240 µL) was added to 10 µL of the urine samples, controls and standards, and after mixing, 200 µL was transferred into a glass vial for injection into the UPLC-MS/MS (Ultra Performance Liquid Chromatography Mass Spectrometry). 

#### 2.2.3. Instrumentation 

The LC-MS/MS (Liquid Chromatography Mass Spectrometry) system consisted of an Acquity UPLC system interfaced with a triple quadrupole mass spectrometer (Xevo TQS-Micro, Waters, Milford, MA, USA) equipped with an electrospray ion source. 

#### 2.2.4. Chromatographic Conditions 

The UPLC separation was performed using an ACQUITY UPLC BEH Amide 1.7 µm, 2.1 × 50 mm column (Waters Corporation, Milford, MA, USA) operating at a flow rate of 200 µL/min, and eluted with a 4 min linear gradient from 90 to 40% acetonitrile in water (2 mM ammonium formate). The oven temperature was set at 40 °C. The injection volume was 5 µL, and the total analysis time, including 1 min for equilibration of column, was 5 min. 

#### 2.2.5. Mass Spectrometer Conditions 

The ESI (Electrospray Ionization) source operates in negative mode, with a capillary voltage of 2.0 kV and a desolvation temperature of 300 °C. The source of the gas was set as follows: desolvation at 200 L/h and cone at 0 L/h. The collision cell pressure was 3.50 × 10^−3^ mbar. The cone voltage and collision energy settings were established individually for each compound for Selected Reaction Monitoring (SRM) detection. The conditions for the detection of lactulose, mannitol and their internal standards obtained by direct infusion of a standard solution (1 µg/mL) were in line with the UPLC at initial mobile phase conditions [27]. 

### 2.3. Fecal Microbiome Analysis 

For cohort I and II, stool samples were collected at a single timepoint, immediately before the L/M BT, and were stored at −80° until DNA extraction. Frozen stool samples were thawed at room temperature, and the DNA was manually extracted using the QIAmp Fast DNA Stool mini kit (Qiagen, Germany) according to the manufacturer’s instructions. DNA was quantified using the NanoDrop ND-1000. 

#### Targeted-Metagenomics

For each sample, the amplification of the V3-V4 region of the 16S rRNA gene was performed by polymerase chain reaction (PCR) to obtain bacterial amplicon libraries (630 bp), using primers reported in the MiSeq rRNA Amplicon Sequencing protocol (Illumina, San Diego, CA, USA) [28]. Internal PCR contaminations were excluded by using negative controls (no template). Moreover, a defined mixture of microbial standard DNA was used as a positive control for sequencing. The sequencing was performed on an Illumina MiSeqTM platform (Illumina, San Diego, CA, USA), where paired-end reads of 300 base-length were generated.

Trimmomatic v. 0.36 software was used to filter raw sequences for their quality and read length [29], and the ChimeraSlayer tool in QIIME 1.9.1 software was employed to filter chimera sequences [30]. Reads were clustered into Operational Taxonomic Units (OTUs) at 97% identity by UCLUST [31] against the Greengenes 13.8 database [32]. QIIME was used to calculate α- and β-diversity and statistical tests (Mann–Whitney U, Kruskal–Wallis, Benjamini–Hochberg tests) were applied on the OTUs’ relative abundances. 

### 2.4. Self-Evaluation of Urological Symptoms (ACSS/UTISA) Questionnaire 

Generally, the diagnosis of acute uncomplicated cystitis is made based on a history of lower urinary tract symptoms in the absence of vaginal discharge; urine dipstick testing and urine cultures can be used only in particular situations [33]. Considering that the diagnosis of acute uncomplicated cystitis is mainly clinical, several dedicated questionnaires, in particular the Acute Cystitis Symptom Score (ACSS) and the UTI Symptom Assessment (UTISA), have been created and validated as diagnostic methods in many clinical settings [34,35,36]. Given the absence in the literature of a dedicated questionnaire for patients suffering from uncomplicated RC, we used a combined ACSS/UTISA questionnaire for this study. The combined ACSS/UTISA questionnaire (questionnaire N° 2 and 3 in the Appendix A) consists of 13 questions. These questionnaires analyze three aspects of the urological manifestations. First, the “typical symptoms”, which consist of urgency and increased voiding frequency, dysuria, incomplete emptying of the bladder, pelvic pain/discomfort, and lumbar pain. Second, the “atypical symptoms”, and third, the subjective perception of how these symptoms have affected the patient’s quality of life in the last year. For each response, a sub-sheet from 0 to 3 was assigned, according to increasing severity. The patients were considered to have a previous acute uncomplicated cystitis if they exceeded the cutoff value of ≥6 in the “typical symptoms” section. In the cohort of patients with GI symptoms, we tried to evaluate the prevalence of lower urinary tract symptoms. Therefore, we administered a combined ACSS/UTISA questionnaire to all patients in the three cohorts, to ascertain the presence or absence of recurrent urinary pathology. 

### 2.5. Self-Evaluation of GI Symptoms (GSRS) Questionnaire 

The GSRS questionnaire contains 15 questions related to five areas of interest in regard to gastroenterological symptoms such as diarrhea, constipation, abdominal pain, reflux, dyspepsia. In this case, each answer was scored from 0 to 3, representing increased severity [20]. All patients from the three cohorts were given this questionnaire. Furthermore, they were asked to qualify their most frequent stool consistency based on the Bristol stool scale.

### 2.6. Statistical Analyses

The demographic and clinical characteristics of the sample were described through descriptive statistical techniques. The qualitative variables were presented through tables containing absolute values and percentage frequencies. Quantitative variables were summarized through the following measures: minimum, maximum, range, mean and standard deviation. The normality of continuous variables was verified with the Kolmogorov–Smirnov test. The primary objective was evaluated by comparing the values of the lactulose/mannitol urinary excretion ratio in two groups of subjects (cohort I and II). The comparison was calculated with the Student’s T-test if the variables were normally distributed and with the Mann–Whitney test in case of the absence of normality. The prevalence of RC in the gastrointestinal cohort (cohort III) was calculated as the percentage of patients who were reported to suffer from recurrent cystitis.

## 3. Results

We enrolled 16 patients in the RC cohort (cohort I) and 15 healthy controls (cohort II), whose characteristics are summarized in Table 1. Furthermore, we enrolled 238 female patients with gastrointestinal symptoms (III cohort) attending the General Gastroenterology and Breath Test Outpatient Clinic.

The prevalence of recurrent cystitis was 100% in cohort I, 0% in cohort II and 20% in cohort III (Table 1). Among the 16 patients in cohort I, 11 patients (68%) had a final diagnosis of GI disease, in particular IBS, inflammatory bowel disease (IBD), gastroesophageal reflux disease (GERD) and lactose intolerance. No GI disease was observed in healthy controls. 

Finally, among 238 patients enrolled in cohort III, including patients seeking gastroenterological advice for GI symptoms, 116 patients (48.7%) had an established GI diagnosis, in particular IBD, diverticular disease, IBS and lactose intolerance.

No significant differences were found between patients affected by RC in cohort I compared to patients affected by RC in cohort III.

### 3.1. Self-Evaluation of Urological Symptoms (ACSS/UTISA) Questionnaire

All patients from cohort I with RC showed a significantly increased median score for the urological symptomatology questionnaire compared to controls (cohort II), for both typical and atypical symptoms (*p* < 0.005). Furthermore, significant differences were also found in 3 items of the ACSS/UTISA questionnaire, which dealt with daily discomfort, daily activity impairment and impairment of social activities (*p* < 0.05) (Figure 1A,B, respectively). 

### 3.2. Self-Evaluation of GI Symptoms (GSRS) Questionnaire

Overall, 68% of patients from cohort I reported GI symptoms, as shown by an increase in the median values of all the items of the GSRS questionnaire (Figure 2A–C). On average, patients with RC showed more intense symptoms, such as diarrhea, constipation and abdominal pain, than controls. Patients with RC also showed great variability in their stool consistency compared to controls, who reported a more homogeneous consistency of type 3 or 4 on the Bristol Stool Scale (data not shown).

### 3.3. Intestinal Permeability Test

Eighty-eight percent of patients with RC from cohort I showed an increased intestinal permeability, with an average value of 0.05 (*p* < 0.05) compared to controls, who did not exceed the established cut-off of 0.03 (Figure 3A,B). 

### 3.4. Alteration at Breath Testing

No statistically significant alterations among RC (cohort I) and controls (group II) were found at breath test analysis. In fact, the AUC of hydrogen and methane did not show any statistical significance. However, a clear trend towards an increased prevalence of SIBO and alterations in oro-cecal transit time were found in patients with recurrent cystitis compared to controls (*p* = ns, Figure 4A,B).

### 3.5. Gut Microbiota Profiling

Microbiota typing showed a trend toward a reduction in biodiversity, which was greater in patients than in controls, as seen from the graphs of α-diversity (i.e., observed species, CHAO 1 and Shannon indexes) (Figure 5A). The patients tended to cluster in a different way compared to controls (Figure 5B) (*p* = 0.02).

Furthermore, the phyla that was most represented in this distribution was Firmicutes, followed by Verrucomicrobia (Figure 5C, left side). At the genera level, potential markers of dysbiosis in RC seem to belong mainly to the phylum of Firmicutes, such as *Ruminococcus*, *Blautia*, *Veillonella* and *Streptococcus* spp. (Figure 5C, right side), while in terms of species, *Acinetobacter* showed particular abundance.

### 3.6. UTIs Etiology

Urinary tract infections appear to be mainly caused by agents derived from the intestinal microflora. The main representative was *E. coli*, but other widely present species include *Streptococcus agalactiae*, *Enterococcus faecalis*, and to a lesser extent, *Shigella* and *Proteus mirabilis* (Figure 6).

## 4. Discussion

In this pilot study, patients with recurrent UTIs described a wide range of negative emotions related to the burden of experienced symptoms and to their impact on the quality of daily life, as well-described in the ACSS/UTISA questionnaire. We found some correlation with previous research on the experiences of patients with UTIs. A recent qualitative, interview-based study by Grigoryan et al. of German and US participants who experienced uncomplicated UTIs, showed a range of negative effects of UTI symptoms on the daily lives, sleep and relationships of the women involved, along with a feeling of helplessness and dread in the context of recurring infections and treatment failure [37]. Similar to this study and to a previous qualitative interview study by Eriksson et al. [38], our patients described a significant impact of urinary symptoms, such as pollakiuria, urgency, suprapubic pain/discomfort, small involuntary urine leakage or sensation of incomplete emptying of the bladder on their daily-life activities. In particular, they complained about daily discomfort and the consequent compromise and impairment of daily activities and social relationships.

As this pilot study has shown, patients with a previous diagnosis of RC not only experienced a significant increase in the incidence of urinary symptoms, but also, they frequently reported GI symptoms, such as dyspepsia, abdominal pain, bloating, flatulence, diarrhea or constipation (or mixed bowel habits), much more than controls, as evidenced by the scores of the GSRS questionnaire. For example, up to 20% of enrolled patients reported significant diarrheal symptoms. However, in most cases, patients with RC showed only mild to moderate symptom intensity. 

The pathogenesis of urinary infections typically starts with contamination of the periurethral region by pathogen microorganisms of the gut, followed by colonization of the urethra and ascending migration to the bladder [39]. Dysbiosis and increased intestinal permeability could contribute to the onset of extra-intestinal disorders, such as RC. In order to better explore the potential role of gut barrier dysfunction in the pathogenesis of RC, we performed an evaluation of intestinal permeability with the H_2_ Breath Test Lactulose/Mannitol, and subsequently, the metagenomic analysis of gut microbiota in the patient and control groups. These evaluations showed a higher incidence of impaired intestinal permeability. At the same time, breath test results showed a trend towards an increased prevalence of SIBO and alterations in the intestinal transit time in patients compared to controls, suggesting the presence of a certain degree of dysbiosis, even considering that urinary infections are mainly caused by components of the intestinal microbial flora, as emerged from our study.

In regard to a possible underlying pathogenetic explanation to our data, it can be assumed that the impaired intestinal permeability observed in the RC cohort, as well as the presence of a pro-inflammatory gut microbiota, could contribute to the dysregulation of enterocytes, with a reduction in the expression of tight junctions, increase in mucosal permeability and dysregulation of immune cells finally leading to an abnormal inflammatory state, which causes mucosal damage and subsequent translocation of microbial fragments in the inner layers of intestinal barrier. Then, the extraintestinal spaces and the areas next to the intestinal tract, such as the urogenital system, may be colonized by gut-derived bacteria, which finally cause the recurrence of cystitis once they reach the bladder [40,41]. Further studies will be required to test the validity of this hypothesis. Unfortunately, probably due to the relatively low number of enrolled patients, we were not able to identify specific bacteria associated with gut barrier dysfunction in this cohort of patients (data not shown).

Conversely, a significant number of patients with GI symptoms reported lower urinary tract symptoms, when investigated with ACSS/UTISA questionnaire. Approximately 20% of GI patients could be diagnosed with RC, based on the symptoms reported in the questionnaires, with a significant impact on quality of life. However, in our population with RC, we cannot exclude that the alterations of intestinal permeability and gut microbiota may be secondary to concomitant gastrointestinal pathologies. In fact, when patients with RC were investigated by gastroenterologists in our outpatient clinic, most of them could be diagnosed with a definite GI disease: IBD (one patient even had the first diagnosis of Crohn’s disease), IBS, lactose intolerance or functional dyspepsia. This would indicate that a significant number of patients with RC might have a misunderstood gastroenterological disease that would predispose them to infection and colonization of bladder. Unfortunately, due to the low number of enrolled patients, our study did not show a significant correlation between RC and specific gastroenterological disease, but the increased prevalence of gastrointestinal diseases in RC patients could highlight a possible common etiology based on dysbiosis and increased gut permeability. However, further studies with a larger cohort of patients are needed to analyze in more depth the specific gut microbiota signature, which could contribute to both RC and different gastrointestinal diseases.

Furthermore, we should take into account that the increasingly massive use of systemic antibiotics for the occurrence of UTIs contributes to the development and spread of antibiotic-resistant pathogens. This also results in the elimination of protective, beneficial microbial species, causing gut and urinary tract dysbiosis, and finally, the recurrence of cystitis itself, which also predisposes patients to other functional gastrointestinal diseases. In this scenario, it is difficult to understand if gastrointestinal disorders in RC patients are primary disorders or they are triggered from antibiotic-induced dysbiosis. Therefore, it is essential to prevent the occurrence of cystitis rather than to just treat it with repeated antibiotic treatment. Several strategies have been suggested in order to prevent the recurrence of cystitis; however, until now, guidelines do not concur on recommendations regarding this topic. Non-antibiotic preventative strategies include the use of cranberry products, despite the low compliance rate among patients, probiotics, phytotherapeutics, or immunotherapies, such as OM-89, which is a bacterial extract from *E. coli* that stimulates the host immune system to produce cytokines and antibodies against several bacteria species due to sharing similar antigenic structures. Vaginal estrogen, methenamine hippurate and replenishment of the glycosaminoglycan (GAG) layers within the bladder urothelium to reduce bacterial adherence, have also been recommended in order to reduce UTIs recurrence but with variable results [42]. 

Our study showed a strong association between altered intestinal permeability, intestinal dysbiosis, SIBO or other gastroenterological pathologies and the development of recurrent cystitis; this should steer our attention towards new therapeutic strategies for the prevention of RC, such as the reconstitution of the intestinal mucosa integrity or the modulation of gut and urinary microbiota with the use of probiotics or even with fecal microbiota transplantation (FMT). Both these therapeutic strategies should determine the displacement of pathogens by probiotics colonization. Supporting these therapeutic strategies, a pilot study including 11 women with RC compared the incidence of symptomatic, culture-proven, antibiotic-treated UTIs in six months pre-fecal microbiota transplantation with six months post-transplantation. The study showed a decrease in symptomatic UTIs after FMT, though not in a statistically significant way [43]. By modulating the microbiome profiles of recipients, FMT could be an innovative therapeutic strategy for refractory recurrent UTI patients, particularly those with antibiotic resistance. Moreover, lactic acid bacteria seem to interfere with the growth and adhesion of urinary pathogens. Therefore, it is necessary to design new dedicated clinical trials to evaluate, in a deeper way, the efficacy of both probiotics - particularly the most effective candidates *L. crispatus* [44], *L. rhamnosus* GR1 or *reuteri* RC14 [45]-and FMT for the treatment and prevention of RC. 

Together with the relatively limited number of patients in this trial, our work has another significant limitation: we have limited information about the nutritional characteristics of the enrolled patients. Given the importance of nutrition in modulating gut microbiota and intestinal permeability, its role in causing urinary infections or improving them should be considered with dedicated trials.

## 5. Conclusions

Patients with RC showed a high prevalence of gastro-intestinal disorders, increased permeability and associated dysbiosis in the microbiota analysis. These results constitute the rationale for further studies to evaluate the potential clinical effects of active gut microbiota modulation on the recurrence of cystitis.

This pilot study laid the foundations for further investigations, and aimed to understand the role of intestinal barrier integrity in greater depth, as its altered permeability appears to be associated with not only intestinal, but also extra-intestinal diseases. Finally, it should be pointed out that various drugs, such as antibiotics, probiotics, prebiotics, a specific diet and numerous pathological conditions could influence the permeability of the intestine through the modulation of the microbial composition. In this context, the study of the degree of intestinal permeability and the potential role of microbiota modulation using new, reliable, reproducible, non-invasive methods could become a valid diagnostic and therapeutic tool for the clinician, thus allowing the development of new and increasingly personalized therapies. 

## Figures and Tables

**Figure 1 jpm-12-01005-f001:**
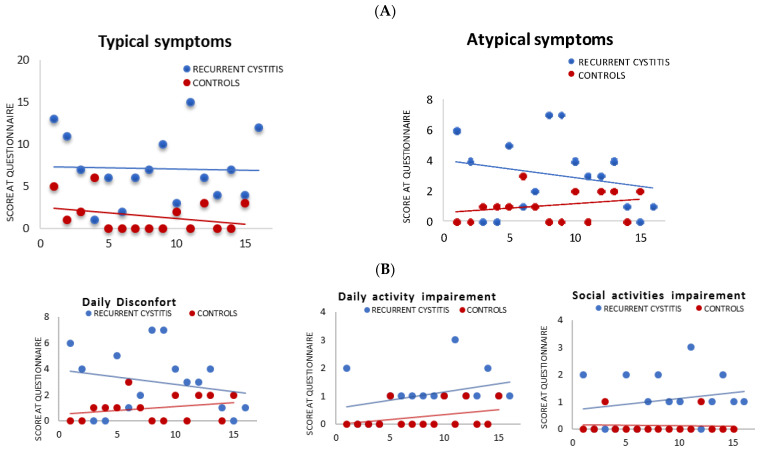
Symptoms and quality of life impairment in patients affected by RC (cohort I): ACSS/UTISA scores. (**A**) ACUTE CYSTITIS SYMPTOM SCORE (ACSS) AND UTI SYMPTOM ASSESSMENT (UTISA); (**A**) SELECTED ITEMS FROM ACSS/UTISA QUESTIONNAIRE. Patients with recurrent cystitis showed higher scores for the questionnaires compared to controls, both in the area of typical and atypical symptoms (**A**) Similarly, they scored significantly higher in items evaluating the perceived impact on quality of life. Selected items are reported in (**B**). All the differences were statistically significant (*p* < 0.05).

**Figure 2 jpm-12-01005-f002:**
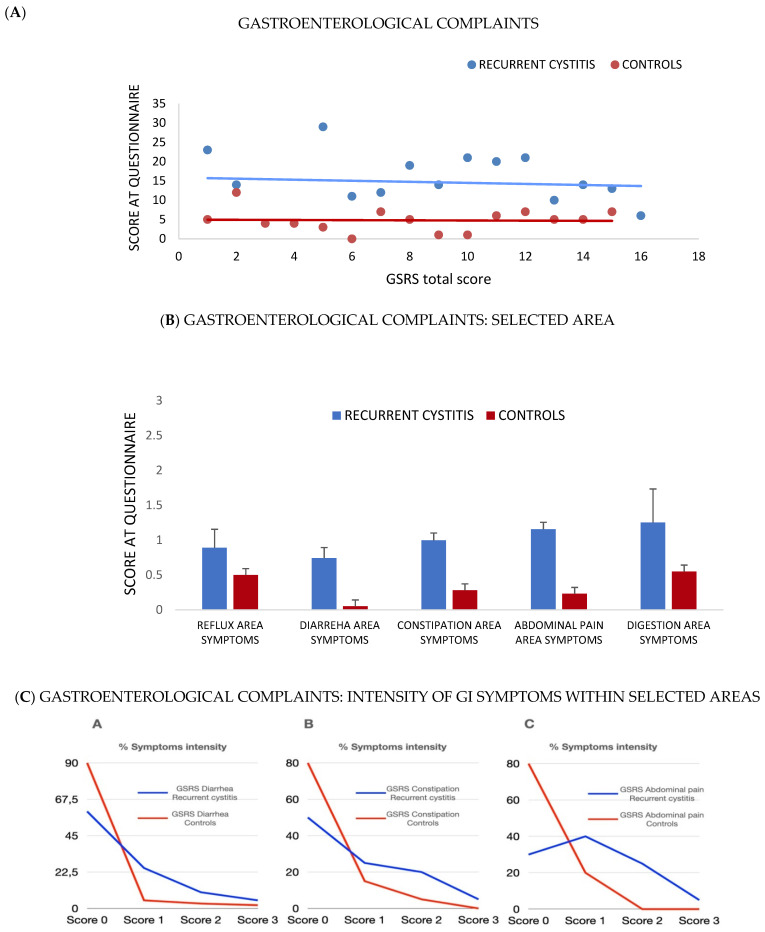
Gastroenterological complaints in RC patients (cohort I): Gastrointestinal Symptoms Rating Scale (GSRS) score. This questionnaire contains 15 questions related to five areas of interest in the gastroenterological clinic, concerning symptoms such as diarrhea, constipation, abdominal pain, reflux, dyspepsia. Among the enrolled patients, most showed an increased prevalence of all the items in the GSRS questionnaire. (**A**) Full GSRS score. (**B**) GSRS SCORE divided per gastrointestinal symptoms area score: reflux, diarrhea, constipation, abdominal pain, indigestion. (**C**) Intensity of symptoms according at GSRS for each single area: percentage of patients indicating a score of 3 (higher) or 2 or 1 or 0 (lower), respectively. Higher scores are consistent with increased severity of symptoms.

**Figure 3 jpm-12-01005-f003:**
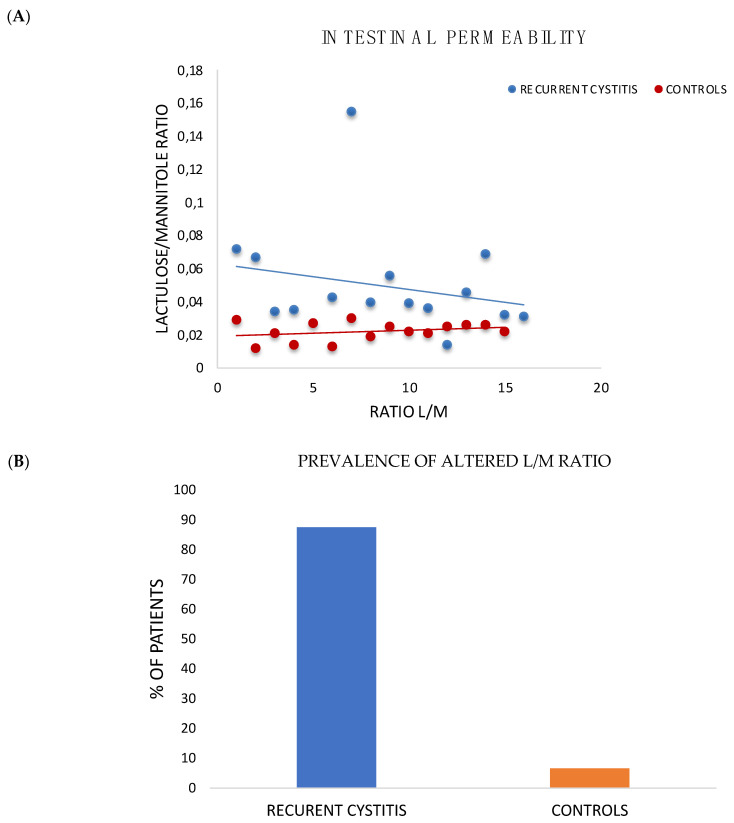
Intestinal permeability modification in patients with RC (cohort I). (**A**) Intestinal permeability. Patients affected by recurrent cystitis showed a statistically significant increase in intestinal permeability, measured as L/M ratio (lactulose/mannitol) with an average urinary ratio of lactulose/mannitol equal to 0.050 compared to 0.02 of controls. (*p* < 0.05). (**B**) Prevalence of altered L/M ratio. Of patients affected by recurrent cystitis, 88% displayed an altered L/M ratio compared to controls.

**Figure 4 jpm-12-01005-f004:**
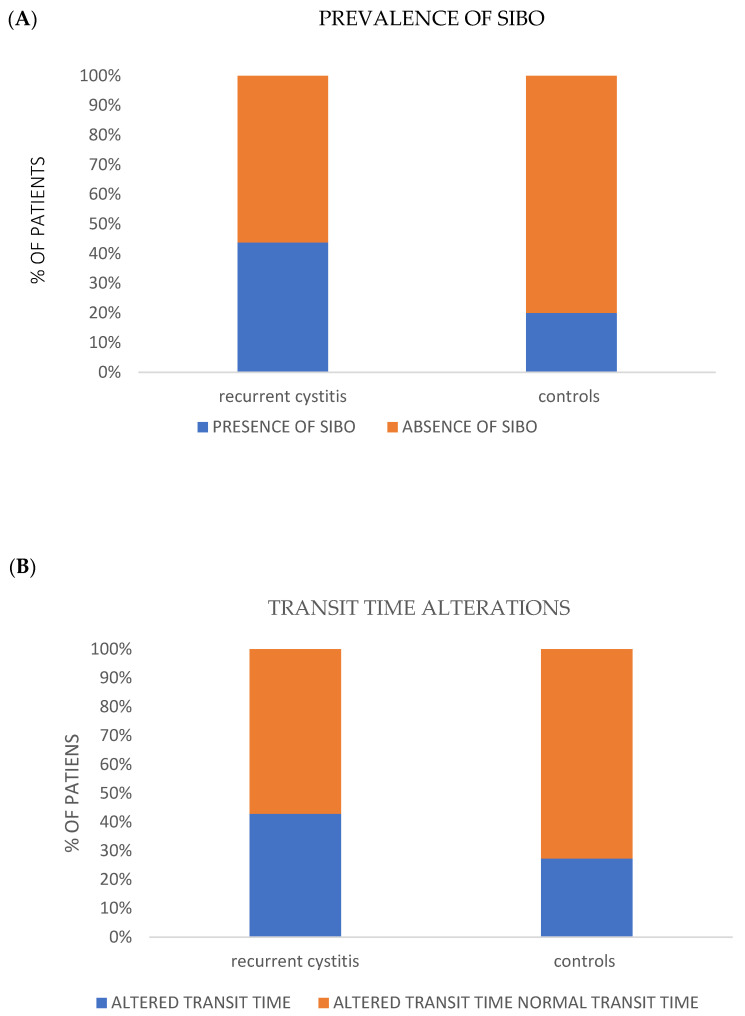
Increased prevalence of alteration at breath testing in RC (cohort I). Prevalence of SIBO (**A**) and prevalence of oro-cecal transit time alterations (**B**). The Breath Test showed that patients with recurrent cystitis showed a trend towards an increased prevalence of SIBO and alterations of the oro-cecal transit time, compared to the control population (differences were not statistically significant, *p* > 0.05).

**Figure 5 jpm-12-01005-f005:**
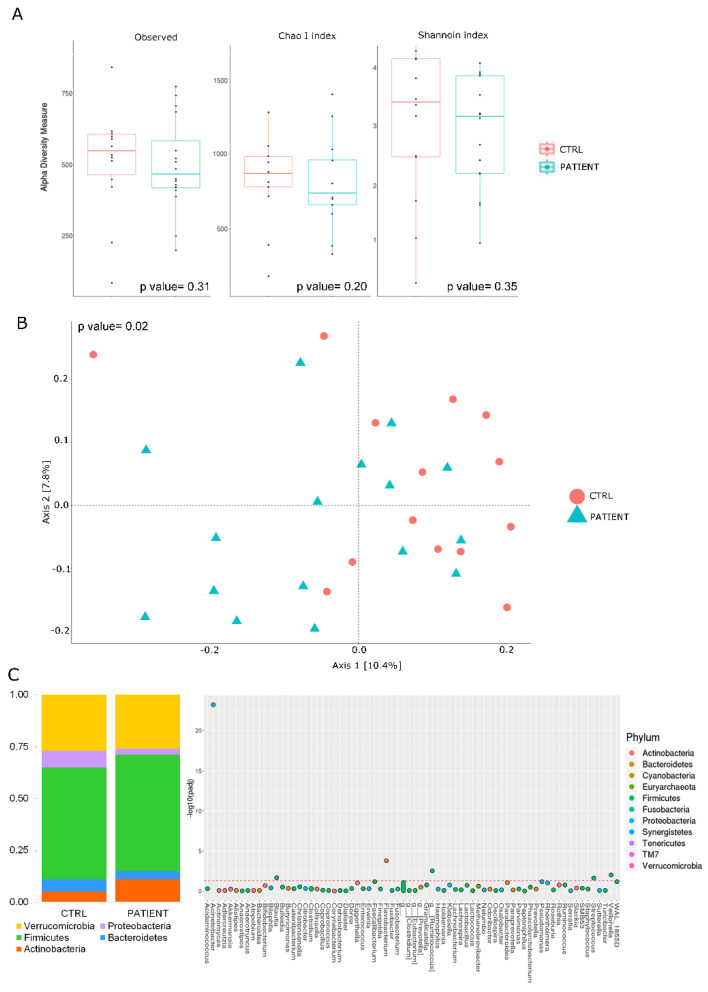
Gut microbiota alteration in RC patient (cohort I). (**A**) Boxplots representing α-diversity indices. The interquartile range is represented by the box and the line in the box is the median. The whiskers indicate the largest and the lowest data points, respectively, while the dots symbolize samples. The analysis of the gut microbiota showed a certain degree of reduction in the observed species and of the CHAO 1 and of Shannon indexes between the two groups. Furthermore, a greater degree of reduction in biodiversity seems more evident in the group of patients (cohort I) versus controls (cohort II). (**B**) *β* diversity analysis performed by Bray Curtis distance matrix and plotted by PCoA plot. Patients affected by RC (green, PTS, cohort I) tend to cluster differently than controls (red, ctr, cohort I). PERMANOVA *p* value = 0.02. (**C**) Phylum distribution (left side) and species distribution between RC patients (cohort I) and controls (cohort II) (right side). Firmicutes and Verrucomicrobia were the most represented phylum of gut microbiota (left side). In terms of prevalent microbial species, some species seem more abundant than others species. In the controls, particular abundance was found for *Acinetobacter*, while the most candidate species as potential markers of dysbiosis in the course of recurrent cystitis seem to belong above all to the phylum of *Firmicutes*, such as *Ruminococcus*, *Blautia*, *Veillonella*, *Streptococcus spp.* Mann–Whitney U test *p* values ≤ 0.05 (right side).

**Figure 6 jpm-12-01005-f006:**
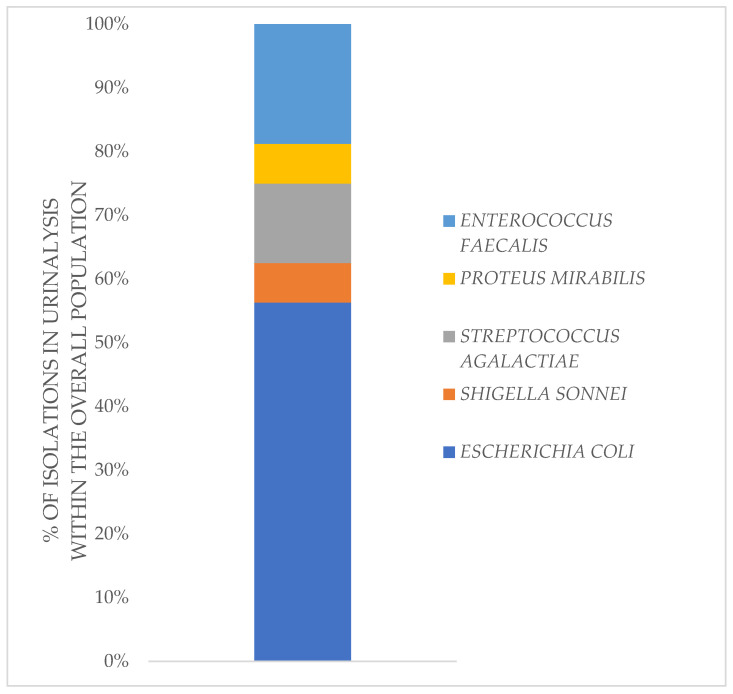
UTIs etiology. Urinary tract infections appear to be mainly caused by agents derived from the intestinal microflora. The main representative was *E. coli*, but other widely present species included *Streptococcus agalactiae*, *Enterococcus faecalis*, and to a lesser extent, *Shigella* and *Proteus mirabilis*.

**Table 1 jpm-12-01005-t001:** The demographic characteristics and medical history of patients in the RC and gastrointestinal cohort and of healthy controls are summarized.

	Patient Affected by Recurrent Cystitis (RC, Cohort I)	Healthy Controls (Cohort II)	Patients Attending GI Outpatient Clinic (Cohort III)
Subjects number (f)	16	15	238
Mean age	44 (+/− 8 years)	42 (+/− 6 years)	42 (+/− 15 years)
Recurrent cystitis prevalence	100%	0%	20.2%
Gastrointestinal diseases prevalence	68% (11/16)	0%	48.7% (116/238)
IBD	18.75% (3/16)	0%	5.04% (122/238)
IBS and chronic functional bowel disorders	37.5% (6/16)	0%	16.4% (39/238)
Dyspepsia/GERD	43.75% (7/16)	0%	8% (19/238)
Lactose intolerance	37.5% (6/16)	0%	19.3% (49/238)
Diverticular disease	0%	0%	2.1% (5/238)

## Data Availability

Data is contained within the article or Appendix A.

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
