# Peer review of "Intestinal Permeability and Dysbiosis in Female Patients with Recurrent Cystitis: A Pilot Study"

_jpm, 2022, doi:10.3390/jpm12061005_

Round 1
Reviewer 1 Report
The present manuscript describes the relationship between intestinal permeability and recurrent urinary tract infections (RC). The manuscript is interesting, however, it presnets with several flaws:
- I am concerned about the low number of RC patients (n=16). Since it is a prevalent disease, could the authors recruit more patients to make more meaningful conclusions?
- Table 1: SD for age is missing in the group III.
- Since the healthy controls have no history of cystitis and UTI, it does not really make sense to compare the differences in the questionnaire scores.
- How was the diagnosis of RC in GI patients established? Just based on the self reported symptoms?
- How was the diagnose of GI diseases established in the 68% of RC patients?
- Please change all bar chart graphs (when reporting on continuous variables) in scattered dot plot graphs.
- The data visualization and choice of graphs is really poor and it should be changed to make the manuscript more attractive and to decipher relevant information.
- Figure 6: Which group of patients/HC is included here?
- Were patient receiving any therapy? This could also influence the microbiome and it should be discussed.
- The major limitation of the study also includes that the conclusions are drawn based on the self-reported symptoms and signs that may be biased and not accurately reflect the disease state.
- Is altered micropbiota a cause or a consequence of RC?
- how was the statistics performed to analyse differences in microbiome between patients and controls?
Author Response
REVIEWERS’ 1 COMMENTS
Comments and Suggestions for Authors:
The present manuscript describes the relationship between intestinal permeability and recurrent urinary tract infections (RC). The manuscript is interesting, however, it presents with several flaws:
- I am concerned about the low number of RC patients (n=16). Since it is a prevalent disease, could the authors recruit more patients to make more meaningful conclusions?
Response: In principle we agree with the reviewer. Unfortunately, we cannot continue the enrollment, as in our hospital some tests were suspended due to the pandemic and are still discontinued: it is the case of the lactulose/mannitol permeability test associated to a Breath Test analysis (up to now we are doing only lactulose/mannitol permeability test without breath test analysis). This major change could make results not perfectly matching. However, based on statistics performed and results obtained, we are positive about association we observed. In the future, when we will be able to perform breath test again we will be glad to confirm these preliminary results in a larger cohort of patients.
- Table 1: SD for age is missing in the group III.
Response: Thank You for pointing out this inaccuracy. Standard deviation from cohort III has been added in Table 1.
- Since the healthy controls have no history of cystitis and UTI, it does not really make sense to compare the differences in the questionnaire scores.
Response: we agree with the reviewer, however the questionnaire was done in order to confirm the validation process also in our population (the questionnaire we gave to each patient is a combination of the two available questionnaire for UTI).
- How was the diagnosis of RC in GI patients established? Just based on the self reported symptoms?
Response: In patients with gastrointestinal symptoms, the diagnosis of recurrent cystitis was made not just by patients history review but also during a medical examination involving also exams review. The definition of UTI followed the current guidelines. The majority of patients were referred from the Gynecology Unit of the Fondazione Policlinico Universitario A. Gemelli IRCCS Hospital.
- How was the diagnosis of GI diseases established in the 68% of RC patients?
Response: The diagnosis of GI problems was established following a gastroenterological visit.
According to the rules of good clinical practice, fecal and hematological analysis, instrumental and imaging assessment and Breath Tests prescribed from the specialist confirmed the diagnosis and helped to build the best therapeutic and personalized approach.
- Please change all bar chart graphs (when reporting on continuous variables) in scattered dot plot graphs.
Response: As required, Figures 1 and 3A have been changed to scattered dot plot graphs.
- The data visualization and choice of graphs is really poor and it should be changed to make the manuscript more attractive and to decipher relevant information.
Response: Thank You for your suggestion. As required in point 6 we changed all bar chart graphs reporting on continuous variables into scattered dot plot graphs. We believe that these new graphs will make our manuscript more attractive and understandable for readers.
- Figure 6: Which group of patients/HC is included here?
Response: In this figure, patients with recurrent cystitis are shown.
- Were patient receiving any therapy? This could also influence the microbiome and it should be discussed.
Response: thanks the reviewer to pointing out it. As described in materials and methods, we enrolled people not taking medicine for at least 1 month before.
- The major limitation of the study also includes that the conclusions are drawn based on the self-reported symptoms and signs that may be biased and not accurately reflect the disease state.
Response: The diagnosis of recurrent cystitis was performed on physical examination, on the patient's declaration of having had 2 episodes of cystitis in the last 6 months or 3 events in the last year, on clinical symptoms such as pollakiuria, dysuria, haematuria and on urine culture with antibiogram. Therefore, the recurrent cystitis diagnosis was made on evaluation of the culture of urine samples based on clinical suspicion.
- Is altered microbiota a cause or a consequence of RC?
Response: This is yet to be defined. This pilot study aims to evaluate the presence of gut microbiome dysbiosis and intestinal permeability alteration in female patients with RC. Furthermore, we wanted to evaluate the possible relationship between impaired gut barrier function and RC. In addition, the crucial role of intestinal barrier integrity and as its altered permeability appears to be associated with not only intestinal but also extra-intestinal diseases.
- how was the statistics performed to analyse differences in microbiome between patients and controls?
Response: The statistical tests applied for the OTUs' comparison between patients and CTRL were the Mann-Whitney U test, and the Benjamini-Hochberg test for the p value correction, as reported in materials and methods and in the legend of figure 5.

Reviewer 2 Report
Summary
The manuscript focuses on gut microbiome dysbiosis and intestinal permeability in 16 female patients with recurrent cystitis compared to 15 healthy females and 238 patients with chronic gastrointestinal disorders. The results showed high intestinal permeability with lowered biodiversity of gut microbiota which indicate that gut barrier impairment severely impacts recurrent cystitis pathogenesis.
Strengths
The study involved cohorts of patients and focuses on a disease that is frequently prevalent in females.
It generated a new method for intestinal permeability index analysis.
Elegant technology like targeted metagenomics and mass spectrometry were used.
New insights into RC pathogenesis and connections to microbiome were revealed.
Weaknesses
Abstract 1st sentence line 30 – “common” – what is the % or number of patients in the study country or world?
Introduction – the ending needs to include 1 or 2 sentences on the key findings.
Results – Inconsistency in fonts and formatting of the figure labels.
Figure legend titles need to summarize the results of a figure, not describe the figure.
Discussion and Conclusion are written well, but can be improved if 1 or 2 relevant clinicals can be discussed briefly to support the study.
Author Response
REVIEWERS’ 2 COMMENTS
The title of this article is “Intestinal permeability and dysbiosis in female patients with recurrent cystitis: a pilot study”. This is an interesting topic, and it is an area that really needs our attention. However, there are still some areas of the article that need to be revised.
- In the "Materials and methods" section of the article, the author needs to organize the content of this section appropriately, such as adjusting the layout of the public notice to make the article look more concise.
Response: we need to apologize with the reviewer as we cannot get completely the question.
- Figure 1. Symptoms and quality of life impairment in patients affected by RC. In this part, the author may add more in-depth discussion and compare their results with the manuscripts recently published in authoritative journals.
Response: We thank you for this precious comment which let make our manuscript more interesting and complete for the readers. We edited the paper (lines 518-533) discussing more in-depth the psychological implications and the quality-of-life impairment caused by RC in our patients.
- In the "Discussion" section of the article, the authors mention the relationship between the pathogenesis of RC and intestinal pathogenic microorganisms, for which the authors need to discuss in more depth and give more of their own opinions and suggestions for the future prevention of RC.
Response: We thank you for your generous suggestion about this fundamental aspect. We deeply analyzed the current state of art about possible prevention strategies for the recurrence of cystitis and added our personal opinions and suggestions for the future prevention of RC in lines 591-627.
- The article mentions that RC patients mostly have concomitant gastrointestinal diseases and gives an analysis that gastrointestinal diseases may have some association with RC. This part can be studied in more depth, such as finding the association of RC with other diseases or studying the composition of the intestinal flora of RC patients as a way to clarify the association between RC and gastrointestinal diseases.
Response: This is a very interesting point, and we thank you for raising it. We added some speculations and personal opinions about this topic in lines 577-590; however, due to the relative low number of enrolled patients, our study does not show significant associations between RC or the composition of gut microbiota of RC patients and specific gastrointestinal diseases. Nevertheless, we believe that this crucial point will deserve future dedicated clinical trials.
- Authors are requested to carefully check the format of the references used in the article to ensure that the references are in the required format.
Response: Thanks for this suggestion. We checked the format of all references used in the article and made them in the required format.

Reviewer 3 Report
The title of this article is “Intestinal permeability and dysbiosis in female patients with recurrent cystitis: a pilot study”. This is an interesting topic, and it is an area that really needs our attention. However, there are still some areas of the article that need to be revised.
1. In the "Materials and methods" section of the article, the author needs to organize the content of this section appropriately, such as adjusting the layout of the public notice to make the article look more concise.
2. Figure 1. Symptoms and quality of life impairment in patients affected by RC. In this part, the author may add more in-depth discussion and compare their results with the manuscripts recently published in authoritative journals.
3. In the "Discussion" section of the article, the authors mention the relationship between the pathogenesis of RC and intestinal pathogenic microorganisms, for which the authors need to discuss in more depth and give more of their own opinions and suggestions for the future prevention of RC.
4. The article mentions that RC patients mostly have concomitant gastrointestinal diseases and gives an analysis that gastrointestinal diseases may have some association with RC. This part can be studied in more depth, such as finding the association of RC with other diseases or studying the composition of the intestinal flora of RC patients as a way to clarify the association between RC and gastrointestinal diseases.
5. Authors are requested to carefully check the format of the references used in the article to ensure that the references are in the required format.
Author Response
N/A
Round 2
Reviewer 1 Report
No further comments.
Reviewer 3 Report
The authors have made corresponding revisions, now the manuscript can be accepted.
This manuscript is a resubmission of an earlier submission. The following is a list of the peer review reports and author responses from that submission.
Round 1
Reviewer 1 Report
Authors in article entitled: "Intestinal permeability and dysbiosis in female patients with recurrent cystitis: a pilot study." evaluated correlations between recurrent cystitis, gut microbiota, intestinal permeability and diseases of gastrontestinal tract. This subject is interesting and authors showed a novel way to investigate intestinal permeability in living subjects.
Broad comments
This is a well-planned research article on interesting and current topic of interactions between host's organism, action of symbiotic microbiota and function of urinary and gastrointestinal system. Unfortunatelly, there are several negative aspects of this study, that require changes in presentation of data and English formating to improve quality of this research article.
Specific comments
1. Introduction section is well-organized, however; additional references would be useful to support presented state of the art.
2. Authors in Introduction on line 53-57 hypothesize about possible anterograde route of infection. Are there any scientific papers on this topic that might support this statement? if yes please support citation.
3. Authors did not write description of collection of samples in subsection entitled "Sample Collection and Treatment" (Page 4 lines 153-158). Only description of laboratory techniques has been provided.
4. In Discussion section authors should more carefully separate what is a result of present research paper and what is hypothesized based on results obtained by others. Specifically in lines 349-356, where authors wrote about results that were not investigated in this project including tight-junction proteins and intestinal inflammation. It is a possible consequence, but should be evaluated and discussed as hypothesis not a fact.
5. Tables should be in one style, authors are encouraged to unify Table 1 and 2 in the same style. (Table 1 should be in style with Table 2).
6. Presentation of data of the study is the main disadvantage of presented project. Results are not clearly presented in Figures.
- Figures 1,2,3,4,5 and 9 should be revised and changed.
- Figures 1 and 2 lack description of statistics. On Figure 1B there is no statisticall significance between groups? Additionally, on Figure 2 what do those dots mean, what kind of significance? there is no description of statistics in caption.
- Figures 5 and 9 should be changed to bar graphs, pie charts are not ideal form for data presentation, and also on Figure 5B there is lacking numbering in distribution of data of green-blue chart.
- Figure 7 requires proper description of beta-diversity between groups, similarly to that presented in Figure 6.
7. Please carefully check all the abbreviations in the text to be sure that they were explained by their first appearance. For example abbrev. explanation is lacking in lines: 34,149, 157, 232 and Tables 1 and 2.
8. English editing is required in the manuscript as a whole. Examples: line 153, 333 (and, subsequently and), 364, 376 (permeability of what?).
9. Please make sure you are using dots not commas while describing statistical data. On page 8 lines 284-285 there should be p<0.05, and 0.03 not 0,05 and 0,03.
Reviewer 2 Report
The manuscript “ Intestinal permeability and dysbiosis in female patients with recurrent cystitis: a pilot study” is interesting and clearly written. I have minor suggestions to include:
- From where and how the control group was recruited, need to provide details.
- It would be good to add a list of questionnaires used for the study in the supplemental data.
- Add all demographic details, smoking, and alcoholic condition which could be a confounding factor to draw a clear conclusion.
- Age data should be provided SD/SEM in the respective table. Is there any statistical difference between the age of cases and control?
Round 2
Reviewer 1 Report
Authors in revised version of the article entitled: "Intestinal permeability and dysbiosis in female patients with recurrent cystitis: a pilot study." did not implement significantly reviewer's suggestions.
- state of the art wasn't extended sufficiently to show scientific background of evaluated topic
- Tables and Figures remain unclear and disorganised. Presented data are far from ideal making it difficult for the reader and reviewer to keep up with the story of the article. Authors did not provide proper changes in the resubmitted article, including:
-Headings in Table 1 are not informative and does not distinguish experimental group from controls and from all cohort. There are spelling errors within the Table and its heading.
-Authors did not include statistical significance in Figures.
-In revised version of the article Figures were subdivided in strange way that diminish their clearity and are separated by text and further describe together in one heading. Each figure should be a separate self-sufficent file that shows data in clear and pleasing way.
-Figures headings are not informative. Authors failed to improve description of Figure 6(revised)/Figure 7(first submision). - In Methodology section in line 173 did authors mean consumption?
Despite the interesting topic authors did not provide satisfying improvement on presented manuscript that still affects significantly its scientific value. Unfortunatelly, I cannot grant aproval for this article to be published.
Author Response
REVIEWERS’ 1 COMMENTS (ROUND 2)
Suggestions for Authors
Authors in revised version of the article entitled: "Intestinal permeability and dysbiosis in female patients with recurrent cystitis: a pilot study." did not implement significantly reviewer's suggestions.
- state of the art wasn't extended sufficiently to show scientific background of evaluated topic
Response: Thanks for your comment. We extended the state of the art in order to show better scientific background of our topic, adding the following references:
- Greenwood-Van Meerveld, B.; Mohammadi, E.; Tyler, K.; Van Gordon, S.; Parker, A.; Towner, R.; Hurst, R. Mechanisms of Visceral Organ Crosstalk: Importance of Alterations in Permeability in Rodent Models. J Urol. 2015, 194, 804-811.
- Ustinova, E.E.; Fraser, M.O.; Pezzone, M.A. Colonic irritation in the rat sensitizes urinary bladder afferents to mechanical and chemical stimuli: an afferent origin of pelvic organ cross-sensitization. Am J Physiol Renal Physiol. 2006, 290, F1478-1487.
- Nickel, J.C.; Tripp, D.A.; Pontari, M.; Moldwin, R.; Mayer, R.; Carr, L.K.; Doggweiler, R.; Yang, C.C.; Mishra, N.; Nordling, J. Interstitial cystitis/painful bladder syndrome and associated medical conditions with an emphasis on irritable bowel syndrome, fibromyalgia and chronic fatigue syndrome. J Urol. 2010, 184,
1358-1363.
- Pezzone, M.A.; Liang, R.; Fraser, M.O. A model of neural cross-talk and irritation in the pelvis: implications for the overlap of chronic pelvic pain disorders. Gastroenterology. 2005, 128, 1953-1964.
- Noronha, R.; Akbarali, H.; Malykhina, A.; Foreman, R.D.; Greenwood-Van Meerveld, B. Changes in urinary bladder smooth muscle function in response to colonic inflammation. Am J Physiol Renal Physiol. 2007, 293, F1461-1467.
- Winnard, K.P.; Dmitrieva, N.; Berkley, K.J. Cross-organ interactions between reproductive, gastrointestinal, and urinary tracts: modulation by estrous stage and involvement of the hypogastric nerve. Am J Physiol Regul Integr Comp Physiol. 2006, 291, R1592-1601.
- Tables and Figures remain unclear and disorganised. Presented data are far from ideal making it difficult for the reader and reviewer to keep up with the story of the article. Authors did not provide proper changes in the resubmitted article, including:
-Headings in Table 1 are not informative and does not distinguish experimental group from controls and from all cohort. There are spelling errors within the Table and its heading.
-Authors did not include statistical significance in Figures.
-In revised version of the article Figures were subdivided in strange way that diminish their clearity and are separated by text and further describe together in one heading. Each figure should be a separate self-sufficent file that shows data in clear and pleasing way.
-Figures headings are not informative. Authors failed to improve description of Figure 6(revised)/Figure 7(first submision).
Response: Thank You for pointing out these inaccuracies. Figures and their respective headings have been revised in order to make the work more fluid and understandable.
We now organized the paper in 6 figures, grouped based on the overall key messages of our findings.
We also modified the current Figure 5 relating to the alteration of the gut microbiota and explained the image exhaustively.
- In Methodology section in line 173 did authors mean consumption?
Response: Yes, in the sentence “during the 6 hours of the test and after assumption of the two sugar solutions: 10 gr of Lactulose and 5 gr of Mannitol” (lines 196-197), the word “assumption” means “consumption”.
